# MSClustering: A Cytoscape Tool for Multi-Level Clustering of Biological Networks

**DOI:** 10.3390/ijms232214240

**Published:** 2022-11-17

**Authors:** Bo-Kai Ge, Geng-Ming Hu, Rex Chen, Chi-Ming Chen

**Affiliations:** 1Department of Physics, National Taiwan Normal University, 88, Sec. 4, Ting-Chou Rd., Taipei 11677, Taiwan; 2School of Computer Science, Carnegie Mellon University, 4665 Forbes Avenue, Pittsburgh, PA 15213, USA

**Keywords:** phylogenetic tree, network visualization, minimum span clustering, Cytoscape tools

## Abstract

MSClustering is an efficient software package for visualizing and analyzing complex networks in Cytoscape. Based on the distance matrix of a network that it takes as input, MSClustering automatically displays the minimum span clustering (MSC) of the network at various characteristic levels. To produce a view of the overall network structure, the app then organizes the multi-level results into an MSC tree. Here, we demonstrate the package’s phylogenetic applications in studying the evolutionary relationships of complex systems, including 63 beta coronaviruses and 197 GPCRs. The validity of MSClustering for large systems has been verified by its clustering of 3481 enzymes. Through an experimental comparison, we show that MSClustering outperforms five different state-of-the-art methods in the efficiency and reliability of their clustering.

## 1. Introduction

Clustering is an important step in studying the present diversity and past evolutionary history of complex systems. For example, Linnaean taxonomy is a hierarchical classification of life on Earth that permits the clustering of related species into clades and phylogenetic trees, allowing the identification of the evolutionary lineages of organisms from their common ancestors [1]. Clustering and classification are both computational approaches for sorting objects into one or more categories based on their features. In general, clustering does not require predefined categories and thus holds enormous potential in its applicability to the modeling of unlabeled data [2,3,4,5].

As many important real-world clustering problems are intrinsically hierarchical, a key desideratum of clustering tools is efficiency in clustering complex systems at various characteristic levels. Typical algorithms are too slow for large datasets and require the number of clusters in the system to be defined a priori. For example, hierarchical clustering is used to cluster an *N*-node system, either using a bottom-up approach or a top-down approach [6]. The standard algorithm has a time complexity of O(*N*^3^) for the bottom-up approach and O(2*^N^*) for an exhaustive search for the top-down approach. In addition to the high computational cost for large datasets, the prerequisite input of the number of clusters is generally unknown for the systems under investigation. To overcome these obstacles, we developed the minimum span clustering (MSC) algorithm for the efficient, automated, and hierarchical clustering of complex systems. Based on the MSC algorithm, we further developed MSClustering, a Cytoscape app to visualize the hierarchical clustering and phylogenetic information of complex systems.

In this work, we demonstrate the validity of MSClustering as a distance-based phylogenetic approach for uncovering the evolutionary relationships between different species and understanding their evolution. In general, the phylogenetic tree of a system can be constructed based on its distances or characters [7]. Character-based methods, such as maximum likelihood [8] or Bayesian inference [9], calculate a score for each tree by considering one character at a time and then optimize the score to derive a phylogenetic tree after comparing all sequences in the alignment. Due to computational costs, an exhaustive search is possible only for small datasets; heuristic searches are implemented for large datasets. Meanwhile, distance-based methods, such as the neighbor-joining (N-J) method [10] or MSC [4], measure the genetic distance between species and construct a phylogenetic tree by linking closely related species together. In this article, we compare the reliability and efficiency of various methods of phylogenetic analysis.

The MSClustering app, which is freely available in Cytoscape [11], enables immediate visualization and statistical analysis of complex systems at various levels, as well as the exporting of publication-quality images from network views in various formats. We link to its user manual and example input files in the Appendix A (part III). A Python version of MSClustering is also available upon request.

## 2. Results and Discussion

We find that MSClustering is generally an efficient and reliable tool for the hierarchical clustering of complex networks. By running MSClustering on an Intel Core i9-9900KF desktop computer, the average hierarchical clustering time (T) for a network of N elements can be approximated by T = 0.0000036 N^2^ + 0.0050 N (s) with R^2^ = 0.992, as shown in Figure 1. For N ≤ 1500, T almost increases linearly with N. The Python version of MSClustering is about 5 times faster, and its computing time can be approximated by T = 0.0000009 N^2^ + 0.0004 N (s) with R^2^ = 0.997. To further demonstrate the performance of MSClustering, we compared it with five other methods to perform the phylogenetic analyses of two biological networks, including a network of 63 beta coronaviruses and a network of 197 GPCRs.

### 2.1. Comparing Phylogenetic Trees of Coronaviruses

We considered a previously studied coronavirus network that contains 63 beta coronaviruses (Appendix A). By inputting the distance matrix from S protein evolution in the WAG model [12], MSClustering provides an automatic clustering of the network at three levels, as shown in Figure 2. Here, blue squares represent nodes at specific levels, and pink squares represent clusters of nodes. These squares are labeled by their sequence IDs or cluster IDs (e.g., “L2G1” denotes “level 2 group 1”). A double-headed arrow represents a core link (i.e., the closest node pair in a cluster), while a single-headed arrow is a non-core link. At level 1, we see clusters of SARS-CoV-2 sequences (L1G1), SARS-CoV sequences (L1G9), and MERS-CoV sequences (L1G8). The virus Bat-SL-RaTG13 (node 4) is a close relative (96.1% nucleotide similarity) of SARS-CoV-2 and is in the same cluster as SARS-CoV-2 sequences (nodes 1–3 and 10) [13]. L1G1 is found to be closely related to L1G2, which contains SARS-CoV-2-like coronaviruses found in pangolins [14]. At level 2, L1G1 and L1G2 merge into group L2G1. L2G4, a group of MERS-CoV sequences, is distinct from other L2 groups containing SARS-CoV (L2G3) and SARS-CoV-2 (L2G1) sequences. Two outliers, L1G6 and L1G12, are detected at level 2. At level 3, we find closer relationships between lineages A and C, and between lineages B and D. From Figure 2, we conclude that this beta-coronavirus network contains four distinct lineages, A, B, C, and D; L1G1 and L1G9 are in lineage B, while L1G8 belongs to lineage C. Our results are consistent with a previous phylogenetic analysis on a similar dataset [15].

By combining results at all three levels, MSClustering builds an MSC tree as shown in Figure 3, where the clustering level of edges is denoted by their color: black for intra-cluster links at level 1, red for intra-cluster links at level 2, yellow for intra-cluster links at level 3, and green for inter-cluster links at level 3. The shortest evolutionary distance between clusters is noted on their connecting edge; the long lengths shown on the green edges imply that there are substantial differences between sequences in different lineages, while sequences in lineage B are evolutionarily closer than those in lineage C.

The MSC tree is consistent with the best character-based phylogenetic tree (Figure 4) created by IQ-TREE [16] based on the S protein evolution. Appendix A shows the AIC and BIC scores of phylogenetic trees constructed with various evolution models. Both metrics suggest that the WAG (+F, empirical AA frequencies; +R4, default free-rate heterogeneity) model is the best empirical maximum likelihood model in this case. The Jaccard similarity in lineage classification is 1.0 between the MSC tree and that predicted by IQ-TREE. Thus, we find that MSClustering enables the inference of past evolutionary events while also delivering information about evolutionary processes based on protein sequences.

### 2.2. Comparing Phylogenetic Trees of GPCRs

To compare the efficiency and reliability of clustering methods, we also performed a phylogenetic analysis of 197 GPCRs as listed in Appendix A. We constructed phylogenetic trees of the GPCRs using three distance-based algorithms (MSClustering, N-J, and the Louvain method) [4,10,17] and three character-based algorithms (IQ-TREE, PhyML, and ProtTest3) [16,18,19]. For distance-based algorithms, the distance is E^d^, where the E-value is calculated by BLAST and d = 1 or d = 0.02. For character-based algorithms, the best substitution model with the lowest AIC is Q.mammal+R6 for IQ-TREE, JTT+R6 for PhyML, and JTT+I+G+F for ProtTest3. Table 1 lists the computation time of these algorithms (without visualization) on an Intel Core i9-9900KF desktop computer. Character-based methods require much more computation time than distance-based methods. From our experience, it is difficult to obtain reliable results with character-based methods for N > 1000 due to the long computation time as well as convergence issues.

The clustering of the GPCRs by MSClustering predicts 47 clusters at level 1, 15 groups at level 2, and 7 classes at level 4 for either d = 1 or d = 0.02; as the level 3 clustering is similar to level 4, we do not consider it here. Figure 5 displays the MSC tree of the 47 level-1 clusters, which combines information regarding their phylogenetic relationships as well as the clustering at levels 2–4. Each level-1 cluster is colored and labeled according to the GPCRdb’s classification [20]. Inter-group edges are noted with their E-value. Detailed results of MSClustering are shown in Appendix A. Specific protein information in Appendix A is obtained from UniProtKB [21]. We find that MSClustering returns the same result for both d = 1 and d = 0.02. At level 4, the system is divided into seven classes, including P2Y receptors (I), P1 receptors (II), prostanoid receptors (III), thyrotropin-releasing hormone receptors (IV), gonadotropin-releasing hormone receptors (V), cannabinoid receptors (VI), and orphan receptors (VII). GPCRdb classifies both P2Y and P1 receptors as nucleotide-like receptors, but they belong to two different groups in the rhodopsin-like family of GPCRs according to comprehensive sequence comparisons and phylogenetic analyses [22]. Our results show that receptors of cannabinoids, thyrotropin-releasing hormones, and gonadotropin-releasing hormones are closer to P1 receptors, while platelet-activating factor receptors are closer to P2Y receptors. At level 2, we find that class II decomposes into groups 5 (A_1_ and A_3_ receptors) and 8 (A_2A_ and A_2B_ receptors). Among the human P1 receptors, the most similar are A1 and A3 receptors (~50% sequence similarity) and A_2A_ and A_2B_ receptors (~60% similarity). Indeed, A_1_ and A_3_ receptors mainly activate the G_i/o_ proteins, which inhibit cAMP production; while A_2A_ and A_2B_ receptors mainly activate G_s_ proteins, which stimulate cAMP production [23]. According to UniProtKB, Nu17, Nu22, Nu29, and Nu30 are clusters of orphan receptors. We find them to be closer to the P1 receptors, but with long distances.

Figure 6 displays the predicted phylogenetic tree of the GPCRs by IQ-TREE with the Q.mammal+R6 model. We find good consistency between the results of MSClustering and IQ-TREE: Each terminal branch of the phylogenetic tree is associated with a level-1 MSC cluster (as labeled), with the structure of branches (as circled by ellipses) being generally consistent with the multi-level MSClustering results shown in Figure 5. The only discrepancies are the positions of Nu17 and Nu30. We find that the closest cluster of Nu17 is Nu15 if the distance is E^d^, but is Nu29 if the distance is derived from the Q.mammal+R6 model. These discrepancies disappear if MSClustering adopts the distance matrix from the Q.mammal+R6 model. Figure 7 displays the phylogenetic trees obtained from PhyML with JTT+R6 (A) and from ProtTest3 with JTT+I+G+F (B). Overall, the tree morphology of Figure 7A is the same as that of Figure 6. In Figure 7B, the four orphan receptor clusters (Nu17, Nu22, Nu29, and Nu30) are located on the side of the P2Y receptors, which is inconsistent with the results of MSClustering, IQ-TREE, and PhyML (where they are on the side of the P1 receptors).

Based on the above comparisons, we conclude that MSClustering is much more efficient than character-based methods and its accuracy is comparable with the best character-based models. For systems with large N, MSClustering can efficiently find a reliable clustering, while character-based methods would have difficulties converging. The validity of MSClustering for large systems can be seen from its clustering of 3481 enzymes in Appendix A (at levels 4–6), where the color of nodes represents the enzyme commission (EC) category. Our results also show that the BLAST E-value provides a good estimation of the evolutionary distance between related proteins.

Figure 8 displays the distance-based clustering results of N-J (A) and Louvain (B) with d = 0.02. Detailed clustering information can be found in Appendix A. The N-J tree of 47 clusters in Figure 8A is roughly consistent with the phylogenetic trees in Figure 5, Figure 6 and Figure 7: Most level-1 clusters are unchanged, six clusters are split (e.g., Th01* denotes the division of Th01 into two clusters), and five clusters are merged (e.g., Nu22/29 contains Nu22 and Nu29). The discrepancy between Figure 6 and Figure 8A can also be seen from the positions of Nu24 and Pl01. The Jaccard similarity between the results of MSClustering and N-J is 0.89 for d = 0.02. We note that N-J’s clustering results are sensitive to monotonic transformations of the defined distance. The N-J trees of the GPCRs are shown in Appendix A for d = 0.02 (A) and d = 1 (B) in a polar layout. The Jaccard similarity between N-J’s results with d = 0.02 and d = 1 is only 0.32. In Figure 8B, we display Louvain’s clustering results of the 47 groups by re-coloring the tree of Figure 6 (since Louvain only provides clustering information). Only P2Y receptors (group 1) and prostanoid receptors (group 2) are properly clustered by Louvain. Group 3 mixes up different types of GPCRs, and groups 4–47 contain only one sequence each. Overall, this suggests that the Louvain method has limited clustering reliability. Based on this comparison, we find that MSClustering outperforms N-J and Louvain in the efficiency and reliability of the clustering that it produces.

### 2.3. Comparing Clustering Plugins in Cytoscape

Table 2 shows a high-level comparison of the five most popular clustering plugins in Cytoscape (AutoAnnotate, clusterMaker2, MCODE, CytoCluster, and ClusterViz) with MSClustering. We include more detailed comparisons in the Appendix A. MSClustering reads the raw distance matrix of a network as an input, while the other plugins can only operate on a previously created and selected Cytoscape network. The first four plugins use multiple algorithms for clustering, while MCODE and MSClustering are implementations of specific clustering algorithms. MSClustering is the only plugin that performs a hierarchical clustering of the network at various characteristic resolutions while not requiring the desired number of clusters to be specified as input. In terms of applications, the MSC tree generated by MSClustering shows specific linkage information at various resolutions and can be applied to phylogenetic studies. Clustermaker2 unifies a variety of algorithms for clustering network attributes as well as for ranking clusters based on potentially orthogonal data. MCODE detects densely-connected regions in a network that can be used to identify potential molecular complexes in protein–protein interaction networks. Finally, the automatic annotation of AutoAnnotate and the GO enrichment analysis of ClusterViz and Cytocluster have potential utility for the investigation of biological networks.

## 3. Materials and Methods

### 3.1. Reading the Input File

MSClustering takes as input the distance matrix of a network and outputs its clustering at various levels. Before running the app, the pairwise distances between network elements must be defined (strict triangle inequality is not required) and calculated. For the protein sequences that we studied in this work, we modeled the distance as the evolutionary distance derived from substitution models or as the BLAST *E*-value. The evolutionary distance denotes the number of substitutions per site that separates a pair of homologous sequences since their divergence from a common ancestral sequence. The *E*-value is a statistic that describes the number of hits expected to be seen by chance when searching for the best-matched region between sequences in a database. A lower *E*-value indicates a more significant match and thus a smaller distance.

To improve the consistency of clustering, two parameters (*m* and *N*_limit_) are required in the input file. The parameter *m* is used for detecting outliers, which are defined as nodes with a distance greater than a threshold *m* × *L*_med_ (*L*_med_ is the median of the shortest distance list). The parameter *N*_limit_ is the minimum number of groups that are desired for the final level of clustering.

### 3.2. Multi-Level Clustering

Based on the MSC algorithm with the flow chart in Appendix A, MSClustering efficiently performs a hierarchical cluster analysis of networks in four steps, as shown in Appendix A: simplification, clustering, renormalization, and outlier detection. We provide a more detailed algorithm description in the Appendix A. In step 1, for a network of *N* nodes, the distance matrix of *N*^2^ elements is converted to the shortest-distance list of length *N*. Since the clustering in step 2 is performed with this simplified list, the clustering speed for large *N* is greatly improved. In step 3, each cluster is considered a node at the next level of clustering, and steps 1 and 2 are repeated until the number of groups is smaller than *N*_limit_. In step 4, to improve the clustering consistency, MSClustering detects outliers using the threshold *m* × *L*_med_ at the last two levels of clustering. Each outlier is considered an individual cluster.

### 3.3. Network Visualization and Analyses

After receiving the distance matrix as input, MSClustering visualizes the clustering at various levels, as shown in Appendix A. It further integrates multi-level results into an MSC tree. In Appendix A, the “Network” panel shows the clustering at three levels along with the MSC tree. For example, the level-2 clustering displays 6 groups, each bounded by a pink square and labeled as L2G*n* (*n* is the group ID). The group L2G2 contains 3 level-1 clusters (L1G3, L1G10, and L1G11), in which the double-headed arrow denotes the group’s shortest link. The arrows between pink squares show the clustering at the next level; unconnected pink squares are outliers. Furthermore, this app is equipped with style-setting tools for preparing publication-quality figures. As shown in Appendix A, the “Style” panel displays options for changing the properties of nodes or edges. Appendix A displays the MSC tree, wherein the clustering level of edges follows the color scheme in Figure 3. Statistical analyses of the resulting networks can be performed with other Cytoscape tools, such as CytoNCA, for calculating network centrality measures [24].

## 4. Conclusions

The MSClustering app is designed to be an easy-to-use Cytoscape tool for visualizing the hierarchical clustering and phylogenetic information of complex systems. Its clustering time is roughly linear in the size (*N*) of the system for *N* < 1500 and is quadratic otherwise. For the coronavirus and GPCR systems that we studied, the MSClustering-generated tree enables the inference of past evolutionary events while also delivering information about evolutionary processes. By comparing MSClustering with five different state-of-the-art methods, we show that it outperforms these methods in the efficiency and reliability of clustering and phylogenetic tree construction.

## Figures and Tables

**Figure 1 ijms-23-14240-f001:**
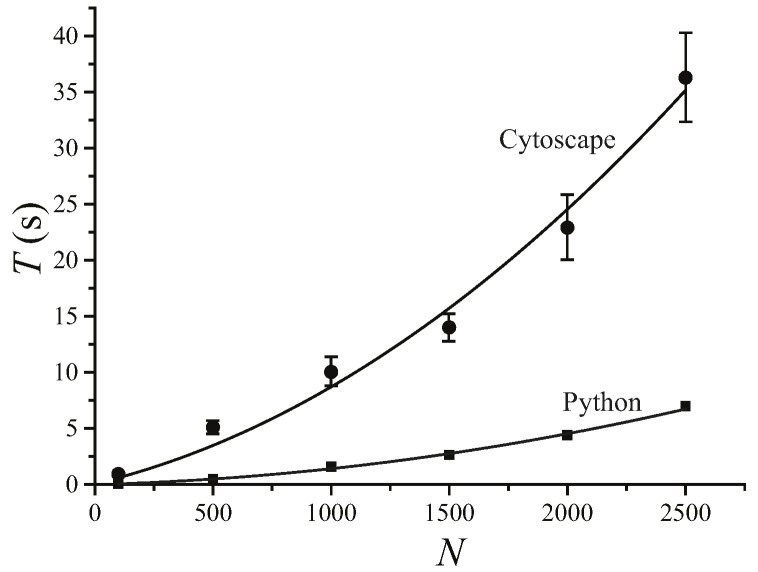
Average computation time (T) of MSClustering in Cytoscape and in Python for the hierarchical clustering of a network of *N* elements on an Intel Core i9-9900KF CPU desktop computer. The solid lines show the curve of best fit.

**Figure 2 ijms-23-14240-f002:**
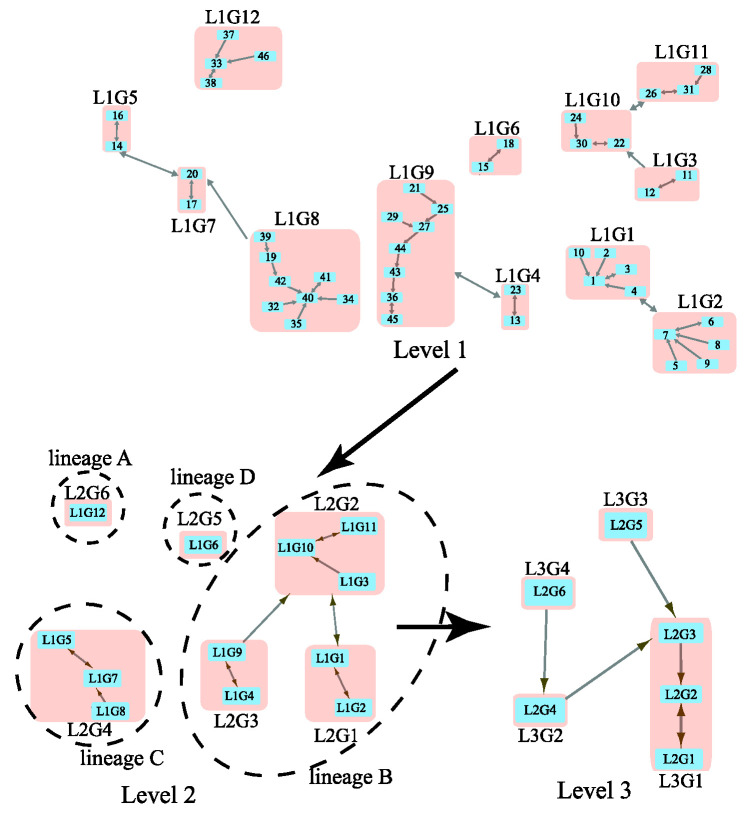
Multi-level clustering of the beta coronavirus network by MSClustering.

**Figure 3 ijms-23-14240-f003:**
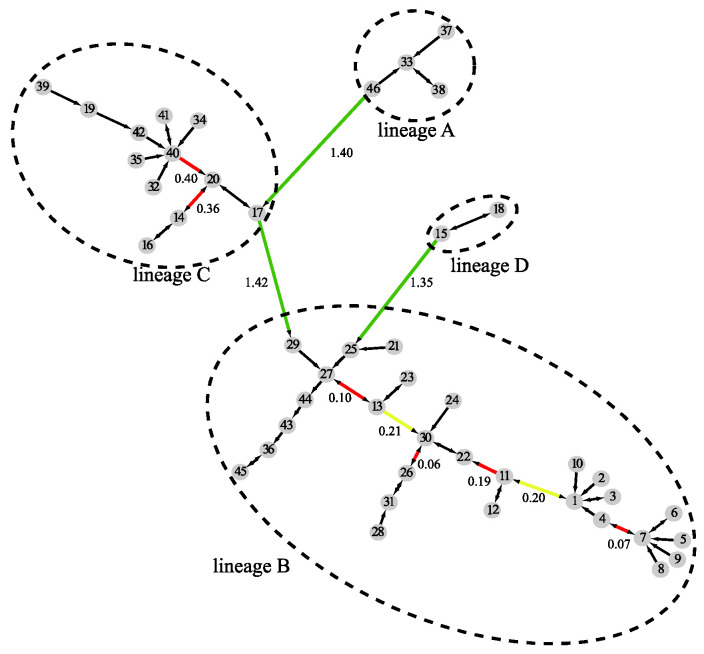
A distance-based phylogenetic tree of the beta coronavirus network, as constructed by MSClustering using the WAG model.

**Figure 4 ijms-23-14240-f004:**
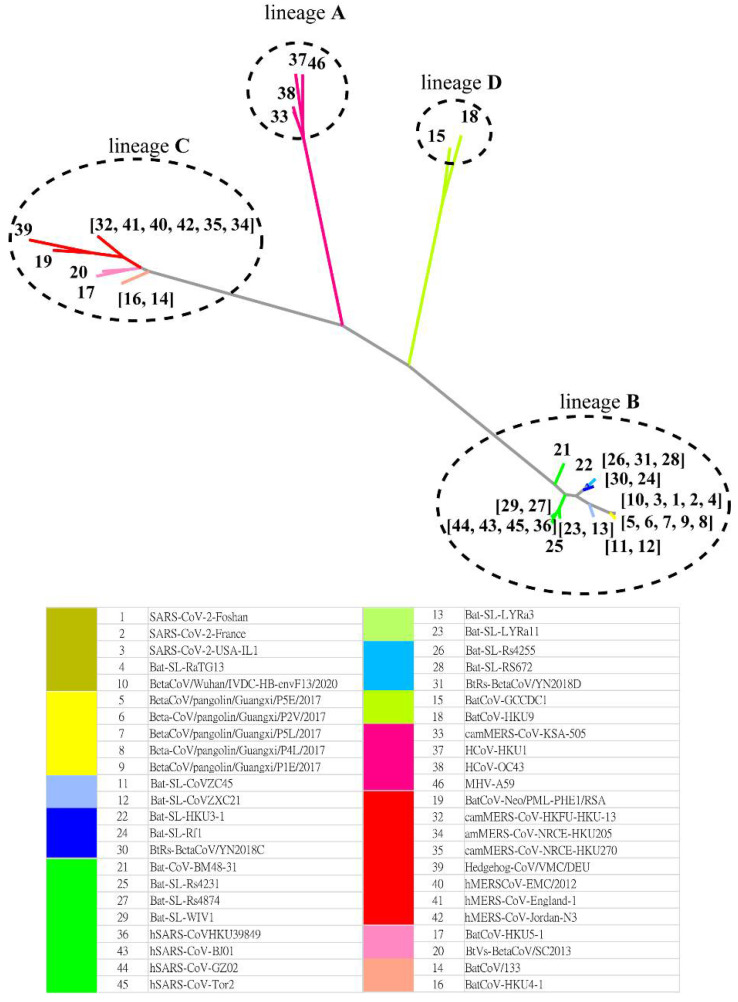
A character-based phylogenetic tree of the beta coronavirus network, as constructed by IQ-TREE with the WAG+F+R4 model. The numbers in the tree diagram refer to the coronavirus sequences in the legend.

**Figure 5 ijms-23-14240-f005:**
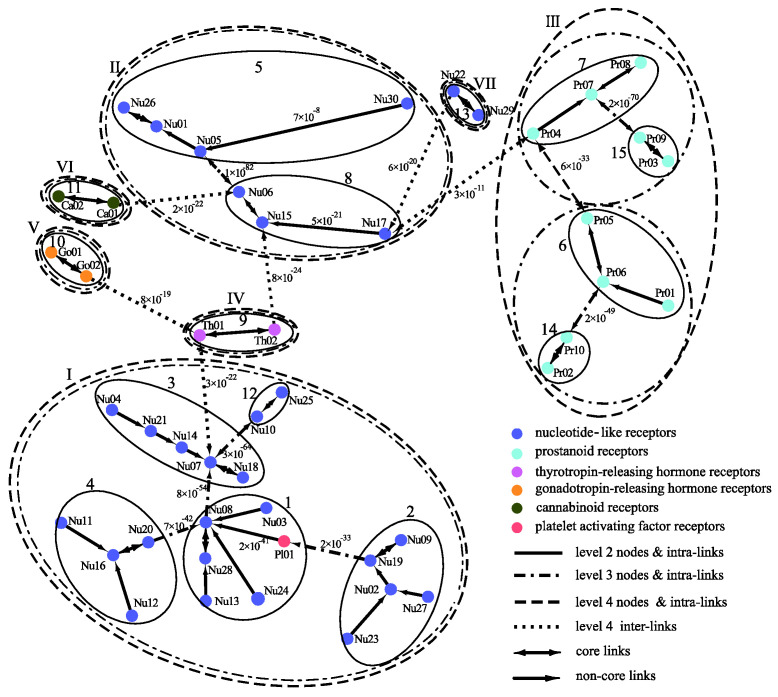
A distance-based phylogenetic tree of the GPCR network, as constructed by MSClustering. The clustering structure of the network is shown for MSC levels 2–4. Each node, colored according to GPCRdb’s classification, is a level 1 MSC cluster. Numerical edge labels show the smallest E values between clusters.

**Figure 6 ijms-23-14240-f006:**
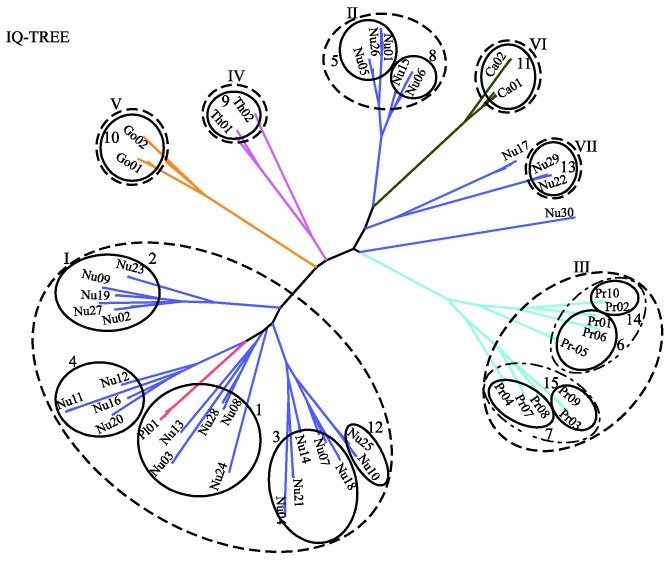
A character-based phylogenetic tree of the GPCR network, as constructed by IQ-TREE using the Q.mammal+R6 model. Each branch is colored according to GPCRdb’s classification.

**Figure 7 ijms-23-14240-f007:**
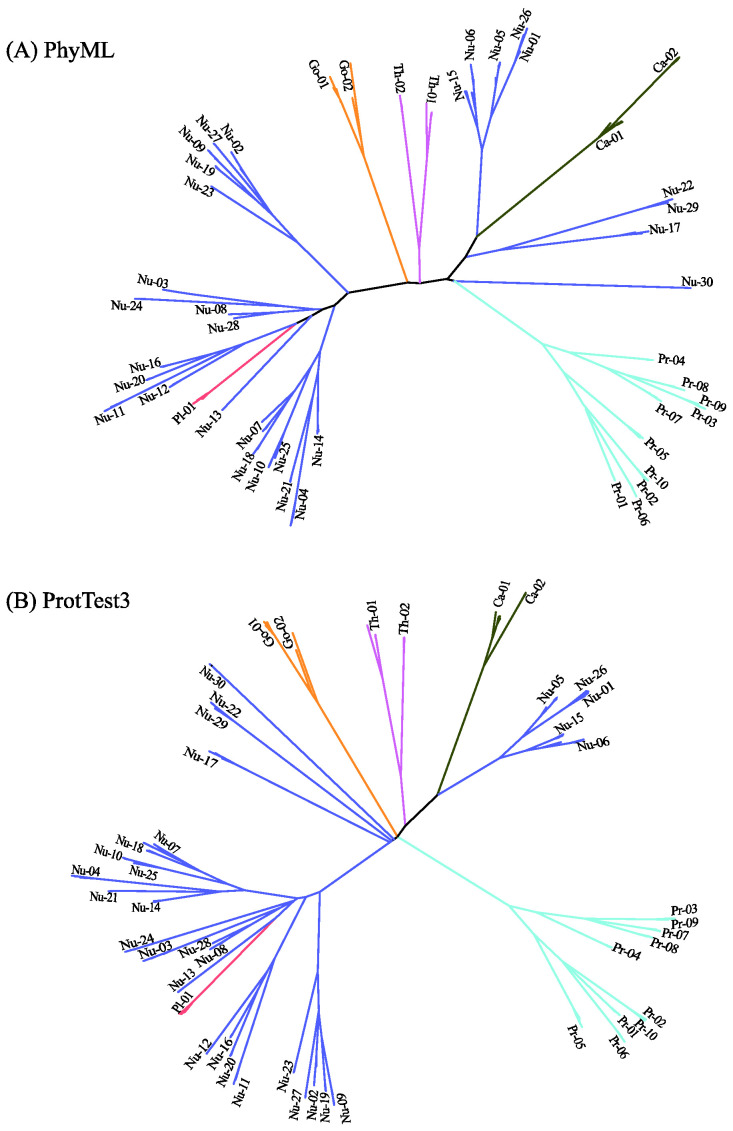
Character-based phylogenetic trees of the GPCR network, as constructed by PhyML using the JTT+R6 model (**A**) and by ProtTest3 using the JTT+I+G+F model (**B**). Each branch is colored according to GPCRdb’s classification.

**Figure 8 ijms-23-14240-f008:**
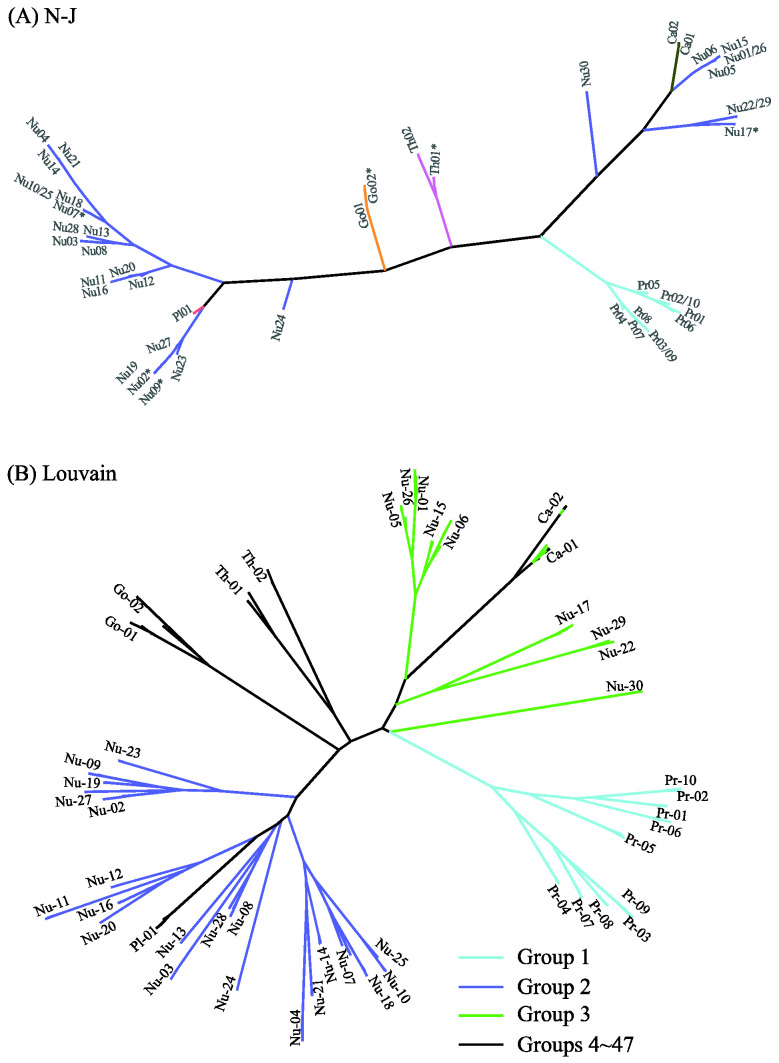
Distance-based phylogenetic trees of the GPCR network, as constructed by the N-J method (**A**) and by the Louvain method (**B**). In (**A**), each branch is colored according to GPCRdb’s classification. The * symbol denotes the splitting of a cluster into two sub-clusters. In (**B**), the first three groups are colored with distinct colors, while groups 4–47 are colored in black.

**Table 1 ijms-23-14240-t001:** Computation time of analyzing 197 GPCRs with distance-based or character-based methods. The first five algorithms were performed on an Intel Core i9-9900KF CPU (3.6 GHz), while ProtTest3 was executed on an Intel(R) Xeon(R) Gold 6154 CPU (3.0 GHz).

	Methods (Without Visualization)	Computation Time (s)
Distance-based	MSClustering	0.0021
	Neighbor-joining	0.036
	Louvain	2.5
Character-based	IQ-TREE	1462
	PhyML	4243
	ProtTest3	29,760

**Table 2 ijms-23-14240-t002:** A high-level comparison of the five most popular clustering plugins in Cytoscape with MSClustering.

Plugin	Features	Clustering Algorithms	Input
AutoAnnotate	finds clusters and visually annotates them with labels and groups	MCL, AP, CF, CC, CCC, and SCPS	selected Cytoscape network
clusterMaker2	unifies a variety of algorithms for clustering networks and attributes as well as for ranking clusters based on potentially orthogonal data	AP and 20 others	selected Cytoscape network
ClusterViz	found cluster can be subjected to GO enrichment analysis	FAG-EC, MCODE, and EAGLE	selected Cytoscape network
CytoCluster	analyzes and visualizes clusters from a selected Cytoscape network	HC-PIN, DCU, IPCA, OH-PIN, IPC-MCE, and ClusterONE	selected Cytoscape network
MCODE	clusters a given network based on the topology to find densely connected regions	MCODE	selected Cytoscape network
MSClustering	an efficient app for hierarchical clustering and phylogenetics of large complex networks	MSC	distance matrix

## Data Availability

The authors confirm that the data supporting the findings of this study are available at BioStudies.

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
