# Peer review of "MSClustering: A Cytoscape Tool for Multi-Level Clustering of Biological Networks"

_ijms, 2022, doi:10.3390/ijms232214240_

Round 1

Reviewer 1 Report

The authors contribute a valuable addition to the tool chest of clustering algorithms. Furthermore, they maximize its utility by publishing an easy to use Cytoscape app.

I have no major or minor critiques.

Author Response

We thank the reviewer for the positive comments. Since the reviewer has no critiques in the report, we have no response to the reviewer's comments.

Reviewer 2 Report

In this manuscript, authors have developed a software package (MSClustering) for visualization and analysis of complex networks in Cytoscape. Authors also demonstrated the software’s utility in studying the evolutionary relationships of complex systems. Performance of MSClustering was evaluated by comparing it with five other methods to perform the phylogenetic analyses of beta coronaviruses and GPCRs network.

The analyses have been well performed and the manuscript is also well written. I believe that the article may be published as it is after checking for typos and grammatical mistakes.  

Author Response

As suggested by the reviewer, we have checked typos and grammatic errors in the revised manuscript.

Reviewer 3 Report

The manuscript titled "MSClustering: A Cytoscape tool for multilevel clustering of biological networks" is seems to a scientifically sound works. Author declare that MSClustering is an efficient software package for visualizing and analyzing complex networks in Cytoscape. However, before recommending this manuscript, i have some major concern,

(1) There are some apps already available in cytoscape for clustering and networking. So, authors must provide a comaparative table with other plugins.

(2) Authors must include these comparision in discussion section also. 

Round 2

Reviewer 3 Report

Recommended for publication